# Physical Properties of *Retisol* under Secondary Pulp and Paper Sludge Application

Marina Butylkina [1] and Elena Ikkonen [2,*]

1    Faculty of Soil Science, Lomonosov Moscow State University, Moscow 119991, Russia; soil.msu@mail.ru
2    Institute of Biology of the Karelian Research Centre, Russian Academy of Sciences, Petrozavodsk 185910, Russia
*    Correspondence: biology@krc.karelia.ru

**Abstract:** A positive effect of pulp and paper mill sludges as a rich source of organic substrates on soil properties was previously found for some types of sludge and soil. In this study, the effect of secondary pulp and paper sludge on water characteristics and thermal properties of *Retisol*, as well as the growth parameters of lettuce (*Lactuca sativa* L.), was tested on the basis of a pot experiment when watering plants with a 20 or 40% sludge solution. The sludge application enhanced plant growth with an increase in biomass accumulation of 21 and 53%, respectively, for 20 and 40% sludge treatments. When the sludge dose was increased from 0 to 40%, the number of leaves increased by 25%, and the leaf mass per area value increased by 42%. Due to the accumulation of more biomass in the shoots than in the roots, sludge causes a change in the allocation of plant biomass. A significant effect of the sludge application on soil particle and microaggregate sized compositions, as well as on the saturated soil hydraulic conductivity, was not found in this study. However, fitted soil water retention curves showed an increased soil water content in sludge-treated soil at all water content values exceeding field capacity. Secondary sludge application led to an increase in the saturated water content from 0.50 to 0.56 $cm^3$ $cm^{-3}$. The 40% sludge solution increased soil thermal conductivity from 0.92 to 0.98 W $m^{-1}$ $K^{-1}$ under saturated water content and from 0.83 to 0.92 W $m^{-1}$ $K^{-1}$ under field capacity. The thermal conductivity was higher in the sludge-treated than untreated soil due to a more pronounced positive effect of increased saturated water content than the negative effect of the increased organic matter content on heat transfer. The positive impact of secondary sludge application on both plant growth parameters and physical properties of *Retisol*, such as increased soil water-holding capacity and thermal conductivity coefficient confirms the possibility of using it to improve soil characteristics and plant productivity.

**Keywords:** *Lactuca sativa*; biomass; water retention curve; hydraulic conductivity; thermal conductivity





## 1. Introduction

Crop growth and yield largely depend on soil properties, such as soil structure and aggregate, water-holding capacty, and aeration [1]. Soil structure and texture play a major role in the infiltration and retention of soil water and, therefore, in plant root distribution and nutrient uptake [2]. Unfavorable physical properties of soils, such as high soil density, low water-holding capacity, low oxygen availability, as well as low soil fertility, limit the crop yield from reaching their productivity potential. Currently, there is a growing need for methods and technologies for improving the properties of agricultural soils, especially soils with poor health and fertility [3,4]. The application of inorganic and organic carbon compounds is widely considered one of the effective methods to improve soil function and quality [5–7]. The positive effect of organic carbon on the stability of soil aggregates, soil cation exchange capacity, and water-holding capacity is widely studied and discussed [8,9].

Pulp and paper mill sludges, generated from the wastewater treatments of the pulp and paper industry, contain a large amount of organic matter and low concentrations of

trace elements and organic pollutants [10]. The positive effect of pulp and paper sludge as a rich source of organic substrates for agricultural soils on crop yield and soil properties was found for different soil types, plant species, and climatic zones [10–15]. In addition to increasing the soil organic matter, the agricultural use of pulp and paper sludges has been found to have positive effects on soil nutrient content, microbial and fauna populations and activity, soil ability to reduce plant disease, as well as on immobilization of toxic elements [12,16–20]. As a result of the organic matter content of paper sludges, the potential benefits to soil conditions, such as porosity, acidification, structural stability, density, and water retention have been shown by early studies [21–23]. Pulp and paper mill sludges application led, as a rule, to a decrease in the soil bulk density [21] and to an increase in the ratio of macropores to micropores [22]. Sludge-mediated macroaggregate formation may be responsible for the improving soil water infiltration and storage [22], thereby providing increased water availability to plants and, consequently, increased plant productivity and yield.

Crop growth and productivity are largely dependent on the pulp and paper sludges application rate [19,24,25] and/or sludge type [10,15,26]. Paper sludge is mainly classified as primary and secondary sludge, with a large proportion of a mixture of primary and secondary sludge. In terms of chemical and physical properties, there are clear distinctions between the sludge types due to an additional biological treatment of primary sludge using microorganisms to reduce the charge in dissolved organic substrates [10,27]. This results in the formation of secondary sludge. To activate biological processes, nitrogen and phosphorus are introduced into biological treatment processes, which leads to an increase in the nitrogen and phosphorus content in the secondary sludge. Secondary sludge has a higher organic content [28], higher phosphorus content, and lower carbon:nitrogen ratio than primary sludge, as well as is generally more difficult to dewater [15]. After the secondary treatment and clarification, water effluents are commonly released into the environment [10].

The beneficial uses and effects of pulp and paper sludge application in agriculture on plant productivity and soil properties have been studied mainly for primary, mixed primary, and secondary or deinking sludges [10]. Much less information is available, however, about the use of secondary sludge in this regard [19]. This appears to be a result of the lower proportion of secondary sludge in total pulp and paper mill sludge generation [12].

For two Mediterranean agricultural soils, an increase in soil pH, C, N, P, and K concentrations was found under the secondary paper mill sludge application [19]. Also, these authors showed that the secondary sludge caused an increase in N and a decrease in Mn, Zn, and Cu content in wheat [19]. A positive effect of secondary pulp and paper mill sludge on the organic matter content of sandy soil and on the grain yield of corn grown on this soil has been established [29]. According to Camberato et al. [12], the increase in crop yields under the secondary sludge application may be attributable to increased nutrient availability for plants. But also, it was assumed that sludges-mediated increase in crop yields is mainly connected with improved soil water properties, primarily water-holding capacity and then nutrient levels [30]. Nevertheless, not much is known about the impact of secondary sludge on soil physical properties, including soils with poor fertility and health.

Despite the decomposition of organic matter and loss of soil carbon, the effects of the pulp and paper mill sludges on soil carbon stores were noted even several years after the sludges were added to soils [21,31]. Moreover, Zibilske et al. [21] showed that the initial (short-term) effect during the first year may be lower than five years after sludge application, likely due to the time required for organic matter to decompose.

This study aimed to evaluate the short-term effect of secondary pulp and paper mill sludge on the characteristics of *Retisol*, such as soil structure and aggregation, hydraulic and thermal conductivity, soil water-holding capacity, saturated water content, field capacity, and available water capacity. This type of soil is often found in northern latitudes and is used for agricultural purposes. However, *Retisol* is characterized by poor fertility and unfavorable physical properties. Increasing the productivity of plants cultivated on this

type of soil is possible only with the use of technologies that improve soil properties, including physical ones. Our hypothesis is that the secondary sludge application can improve the soil properties of *Retisol* by increasing its water-holding capacity and thermal conductivity. Because sludge-mediated changes in soil properties are expected to enhance plant productivity, in this study we assess whether secondary sludge application would improve growth parameters of lettuce (*Lactuca sativa* L.) plants cultivated on this type of soil. The application of secondary sludge into the soil was carried out with each watering of plants during a pot culture experiment.

## 2. Materials and Methods

### 2.1. Pot Experiment Installation

For *Retisol* collection, the field site of the Korza station located in the northwest of Russia was used. Before the collection, the soil was not fertilized over ten years. According to Gael and Smirnova [32], this soil typically is characterized by low natural fertility with a humus content of 0.5–2.5%, as well as low water-holding capacity. The soil had 1.6 g kg$^{-1}$ of total N, 0.1 g kg$^{-1}$ of available P and K, 0.8 g kg$^{-1}$ of Mg and Ca, and a pH value of 5.46 [33].

After, the collected soil was air-dried, passed through a 2 mm sieve, and fertilized with 150 mg kg$^{-1}$ of N, P, and K. After fertilizing, the soil was well-watered and incubated under 21–23 °C of air temperature and 70–80% of the field capacity for 14 days, and then packed into plastic 0.80 L pots with a soil bulk density of approximately 1.4 g dry mass cm$^{-3}$. Two days before sowing, the soil in each pot was moistened until saturated water content.

The seeds of lettuce (*Lactuca sativa* L., var. Medvezhje ushko) were sown with six seeds per pot. The lettuce plants were cultivated under controlled conditions with day/night temperatures of 23/20 °C, 16/8 photoperiod, 250 μmol m$^{-2}$ s$^{-1}$ of photosynthetic photon flux density, and 60–70% of air humidity. After plant germination, all pots were divided into three parts and watered every two days. The first part of the plants were watered with distilled water (0% treatment), and the second and third parts were watered with 20% and 40% solutions of the secondary pulp and paper mill sludges (20% and 40% treatment, accordingly). During the 46-day experiment, 0, 139, and 278 mL of the secondary sludge was added to 1 L of soil, accordingly, for 0%, 20%, and 40% treatments. The sludge used in this study was generated during the secondary aeration of wastewater from the Kondopoga pulp and paper mill (Karelia, Russia). The collected sludge was mixed with distilled water to achieve a content equal to 20 and 40%. These secondary sludge concentrations were chosen in a preliminary experiment. Two weeks after sowing, the seedlings were thinned to one per pot.

### 2.2. Plant Growth Parameters

Five 46-day-old lettuce plants of each treatment were used for the measurement of plant growth parameters. The total leaf number per plant was counted and shoots were separated from the underground plant part. The total leaf area per plant was determined by leaf scanning and using the program "AreaS". The roots were carefully separated from the soil and washed first with running water and then with distilled water. To determine the dry weight, the collected shoots and roots of the seedlings were dried at 70 °C to constant weight. The shoot weight ratio was calculated as the ratio between shoot weight and total biomass. The root weight ratio was calculated as the ratio between root weight and total biomass. The root:shoot ratio was calculated as the ratio between root and shoot weight. The value of leaf mass per area (LMA) was calculated as the ratio between shoot mass and total leaf area per plant.

### 2.3. Soil Sample Preparation

Undisturbed soil samples from pots were used to determine soil water-holding capacity, saturated hydraulic conductivity, and thermal properties of the soil. To accomplish this, the aired part of the plants was cut off so as not to damage the soil surface. The soil

samples were collected with plastic columns 6 cm high and 4.5 cm in diameter. The bottom of the columns was covered with a nylon membrane. The top surface of each soil sample was the surface formed in the pots by watering the plants during the experiment.

### 2.4. Soil Water Retention Capacity

The water retention capacity of soil samples during the study was determined by the capillary metric method using porous ceramic probes 5 cm long [34]. The soil samples were then saturated from the bottom by placing them in larger plastic containers with water for two days. To prevent evaporation the soil samples were covered with a polyethylene sheet. After the saturation period, the soil samples began to dry, and the measurements were taken in the pF (decimal logarithm of the absolute value of the water pressure) range of 0–2.8.

### 2.5. Saturated Hydraulic Conductivity

The saturated hydraulic conductivity (Ks) reflects the ability of soils to pass water through pore spaces under saturated conditions. The Ks values of undisturbed soil were measured in the laboratory using the constant head method [35]. Before starting the measurement, the soil cores were completely saturated. Throughout the experiment, a constant layer of water on the soil surface was maintained at a level of 1 cm. The time of sequential drainage of every 10 mL of water was measured and Darcy's equation [36] was used to calculate the Ks values.

### 2.6. Soil Thermal Conductivity

The thermal conductivity of soil cores were studied with a TEMPOS analyzer (METER Group, Inc., Pullman, WA, USA) connected to a SH-3 sensor. The measurements were carried out on samples of an undisturbed structure under the condition of saturated water content, soil moisture close to field capacity, and under conditions of air-dried soil.

### 2.7. Soil Particle and Microaggregate Size Distribution

The particle size distribution and water-stable microaggregate composition were determined using the Mastersizer 3000E laser diffraction particle size analyzer (Malvern Instruments Ltd., Malvern, Worcestershire, UK). Previously, the soil of each treatment was crushed with a rubber pestle and sieved with a 1 mm sieve. The preparation of samples for particle size distribution analysis was carried out using a Sonifier S-250D ultrasonic processor (BransonUltrasonics, Danbury, CT, USA) with a dispersion energy of about 450 J mL$^{-1}$ and without removing organic matter. The particle size distribution measured in this way included both mineral (primary particles) and organic particles, which made it possible to determine the proportion of organic particles that appeared in the soil after the sludge application. Before the measuring of the water-stable microaggregate composition, the soil samples were rotated in water for an hour at a rate of 80 rpm.

### 2.8. Statistical Analyses

The means ± SE were determined with five replicates for the growth parameters and soil physical parameters. The least significant difference (LSD) of ANOVA was used to evaluate the significant difference between the means. The difference was accepted as significant at the $p < 0.05$ level. For the statistical test, we used Statistica software (v. 8.0.550.0, StatSoft, Inc., Tulsa, OK, USA). The van Genuchten model [37] was used to approximate the data on the dependence of the soil water content on water potential.

### 3. Results

### 3.1. Plant Growth Parameters

As shown in Table 1, the secondary sludge application significantly affected the lettuce plant's growth. In accordance with the increase in the sludge application rate, the leaf number and plant dry weight increased, although this increase was more pronounced for

the aerial parts of plants than for the roots. The different effects of secondary sludge on the growth of plant organs led to an increase in the ratio of shoot weight to total plant biomass from $0.66 \pm 0.01$ to $0.71 \pm 0.01$ and a decrease in the root:shoot ratio from $0.53 \pm 0.01$ to $0.42 \pm 0.02$. Although the total leaves area tended to be the highest in plants of 40% treatment ($321 \pm 13$ cm$^2$ per plant compared to $289 \pm 39$ cm$^2$ per plant for 0% treated plants), the difference was not statistically significant. Among all treatments, the highest LMA values ($51 \pm 3$ g m$^{-2}$) were found for the plants grown under the largest sludge dose.

**Table 1.** Growth parameters of *L. sativa* grown on *Retisol* watering with the 0, 20, or 40% (0%, 20%, and 40% treatments, accordingly) of secondary pulp and paper mill sludge.

| Parameter | 0% | 20% | 40% | *p* |
|---|---|---|---|---|
| Leaf number plant$^{-1}$ | $13.3 \pm 0.7$ b | $15.8 \pm 1.8$ ab | $16.7 \pm 0.7$ a | ** |
| Shoot dry weight, g plant$^{-1}$ | $1.0 \pm 0.2$ b | $1.3 \pm 0.1$ ab | $1.6 \pm 0.1$ a | ** |
| Root dry weight, g plant$^{-1}$ | $0.51 \pm 0.13$ a | $0.61 \pm 0.07$ a | $0.68 \pm 0.06$ a | ns |
| Total weight, g plant$^{-1}$ | $1.5 \pm 0.4$ b | $1.9 \pm 0.2$ ab | $2.3 \pm 0.2$ a | * |
| Shoot weight ratio | $0.66 \pm 0.01$ b | $0.69 \pm 0.01$ a | $0.71 \pm 0.01$ a | ** |
| Root weight ratio | $0.34 \pm 0.01$ a | $0.31 \pm 0.01$ b | $0.29 \pm 0.01$ b | * |
| Root:Shoot weight ratio | $0.53 \pm 0.01$ a | $0.46 \pm 0.03$ b | $0.42 \pm 0.02$ b | ** |
| Leaf area, cm$^2$ plant$^{-1}$ | $289 \pm 39$ a | $303 \pm 20$ a | $321 \pm 13$ a | ns |
| LMA, g m$^{-2}$ | $36 \pm 3$ b | $44 \pm 4$ ab | $51 \pm 3$ a | * |

LMA, leaf mass per area. Different letters show significant differences between the means. Asterisks denote significance levels: * $p < 0.05$, ** $p < 0.01$; ns, not significant.

### 3.2. Soil Particle and Microaggregate Sized Compositions

The surface of the soil treated with the sludge had a darker color compared to the untreated soil. This suggests that not all sludge particles penetrated into the pore space of the soil, and some of them accumulated on the soil surface. As a result of the incomplete transition of sludge particles deep in the soil, the change in the soil particle composition was insignificant. For the soil under this study, silt (0.002–0.05 mm) had the highest particle size fraction, followed by clay (<0.002 mm) and sand (>0.05) (Table 2). We observed that in the aqueous suspension of secondary pulp and paper sludge, there were twice as many fine clay-sized particles and 7% less silt-sized particles than in the untreated soil. For particles 0.05–1.0 mm in size, no significant differences were found between the studied soil and the secondary sludge. The secondary sludge application at the dose of 40% slightly increased the mean content of clay-sized and silt-sized particles, as well as reduced the content of sand-sized particles, but these changes were not large enough to be statistically significant.

**Table 2.** Textures of secondary pulp and paper sludge and *Retisol* samples watering with the 0, 20, or 40% (0%, 20%, and 40% treatments, accordingly) of secondary pulp and paper sludge.

| Particle Size, mm | Secondary Sludge | Soil Samples | | |
|---|---|---|---|---|
| | | 0% | 20% | 40% |
| <0.002 | $13.6 \pm 0.2$ a | $6.5 \pm 0.1$ b | $6.4 \pm 0.1$ b | $6.8 \pm 0.2$ b |
| 0.002–0.05 | $55.3 \pm 0.6$ b | $62.1 \pm 0.4$ a | $62.6 \pm 0.3$ a | $63.3 \pm 0.7$ a |
| 0.05–1.0 | $30.7 \pm 0.8$ a | $31.0 \pm 0.4$ a | $31.0 \pm 0.4$ a | $29.1 \pm 0.8$ a |

Different letters show significant differences between the means.

Shaking soil samples and aqueous suspension of secondary sludge in water led to the disintegration of non-water-stable aggregates. This made it possible to estimate the size distribution of water-stable microaggregates (Table 3). For the studied soil, the composition of aggregated particles >0.05 mm included a significant proportion of clay-sized (<0.002 mm) and silt-sized (0.002–0.05 mm) fractions. The aqueous suspension of secondary sludge did not contain particles smaller than 0.002 mm. No significant differences in the mean values of the content of water-stable aggregates with sizes of 0.002–0.05 or

0.05–1.0 mm were found between the secondary pulp and paper sludge and the soil used in this study. The secondary sludge application in the soil changed the proportion of water-stable aggregates with an increase in the content of particles larger than 0.05 mm, but this increase was not supported by a statistical test ($p > 0.05$).

**Table 3.** Water-stable aggregates content (%) in secondary pulp and paper sludge and *Retisol* samples watered with the 0, 20, or 40% (0%, 20%, and 40% treatments, accordingly) of secondary pulp and paper sludge.

| Particle Size, mm | Secondary Sludge | Soil Samples | | |
|---|---|---|---|---|
| | | 0% | 20% | 40% |
| <0.002 | 0 b | 1.3 ± 0.0 a | 1.2 ± 0.0 a | 1.2 ± 0.1 a |
| 0.002–0.05 | 39.0 ± 0.9 a | 38.1 ± 0.5 a | 37.1 ± 0.5 a | 37.1 ± 1.3 a |
| 0.05–1.0 | 60.9 ± 0.9 a | 60.5 ± 0.5 a | 61.5 ± 0.5 a | 61.5 ± 1.3 a |

Different letters show significant differences between the means.

### 3.3. Water Retention Capacity

The structure and composition of the soil, as well as the architecture of pores spaces, determine its water-holding capacity. The soil water retention curve is a quantitative characteristic of the water-holding capacity of soils and is widely used to characterize the hydraulic behavior of soil. The water retention curve is the equilibrium relationship between the matrix potential and soil water content. The matrix potential is expressed in pF, which is the decimal logarithm of the absolute value of the water pressure, measured in centimeters. The pF values from 0 to 3.0 correspond to the range from the saturated water content to a state in which the continuity of capillary filling with water is interrupted. For soil samples used in this study, the experimental points and fitted soil water retention curves are shown in Figure 1. No significant differences in the soil water content between the untreated and treated soil samples were found at pF > 2.5. According to Klute et al. [38] and Romano et al. [39], the moisture content at a soil water matric potential of pF = 2.5 is quoted as a field capacity. In this study, the water retention curves began to differ widely between the soil samples at pF < 2.5. The differences between the curves increased with the decreasing pF or increasing soil water content. The curves of the untreated soil samples maintained their lower soil water retention capacity. Consistent with the increase in sludge dose used in this study, the secondary sludge application caused an increase in the soil water content at all pF values lower than 2.5. The saturated water content of the soil samples without the sludge ranged from 0.46 to 0.53 $cm^3$ $cm^{-3}$. The secondary sludge application led to an increase in the saturated water content to an average of 0.56 $cm^3$ $cm^{-3}$. The inflection point on the soil water retention curves reflects the water potential at which air enters the largest pores of the soil. This point, called air-entry value, shifted toward higher pF values for samples without sludge, and toward lower pF for samples with sludge.

### 3.4. Saturated Hydraulic Conductivity

Among all the studied treatments, the soil treated with the 20% sludge solution had the highest value of saturated hydraulic conductivity (Figure 2). Moreover, the secondary sludge application caused an increase in the range of saturated hydraulic conductivity values of the soil.

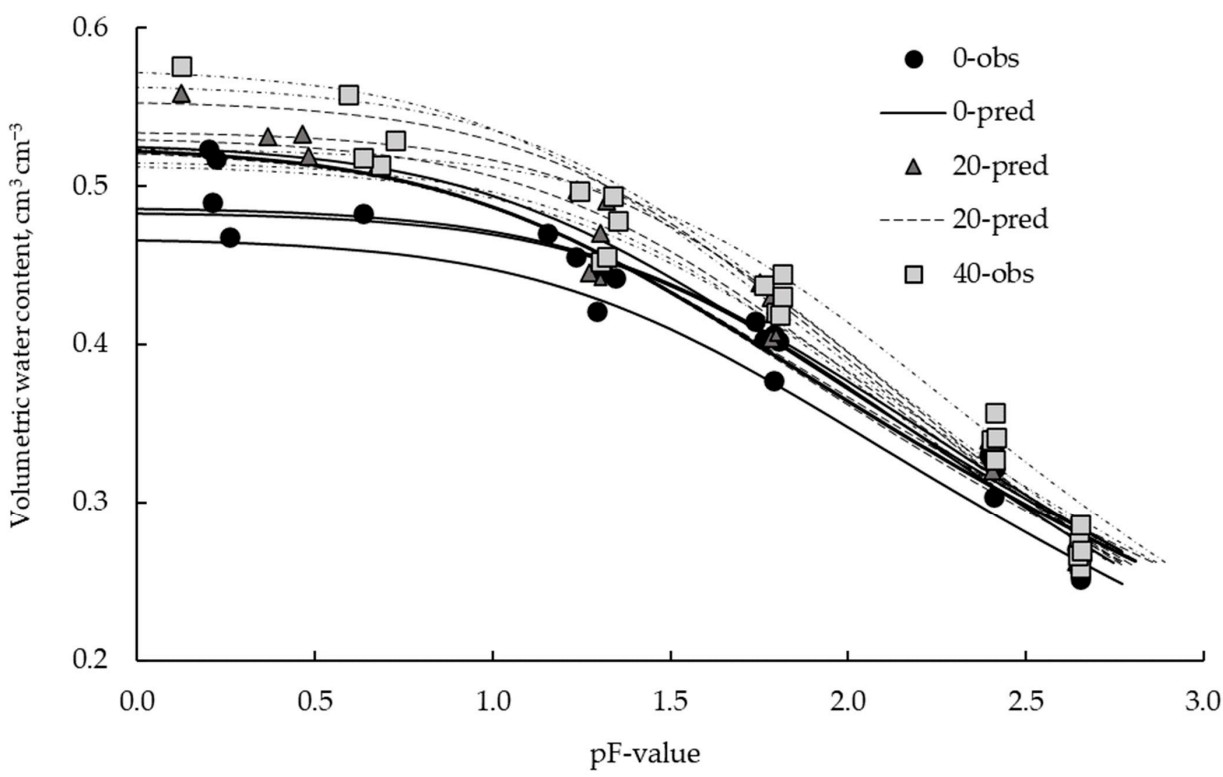

**Figure 1.** Soil water retention curves of *Retisol* samples watered with the 0, 20, or 40% (0-obs (0-pred), 20-obs (20-pred), 40-obs (40-pred), accordingly) of secondary pulp and paper mill sludge. 'obs' means 'observed and 'pred' means predicted. The curves were found for each soil sample.

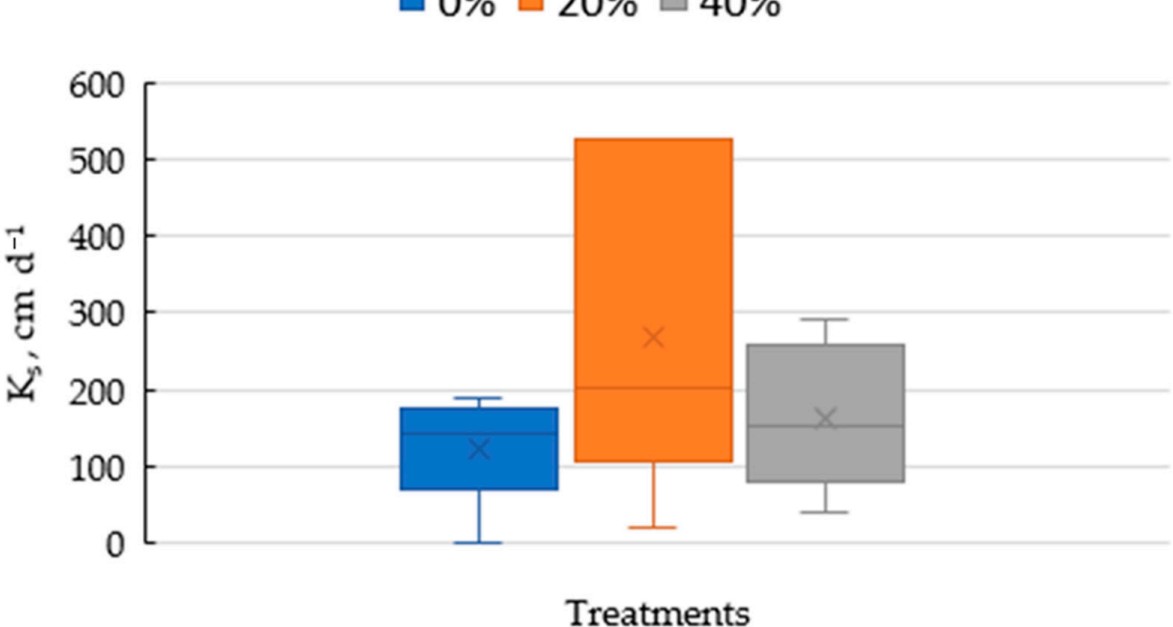

**Figure 2.** Saturated hydraulic conductivity ($K_s$) of *Retisol* samples watered with the 0, 20, or 40% of secondary pulp and paper mill sludge.

*3.5. EffectiveThermalConductivity*

The secondary sludge application with the dose of 40% significantly increased the thermal conductivity coefficient of the studied soil from $0.92 \pm 0.01$ to $0.98 \pm 0.01$ W m$^{-1}$ K$^{-1}$ under the condition of saturated water content and from $0.83 \pm 0.04$ to $0.92 \pm 0.01$ W m$^{-1}$ K$^{-1}$

under the field capacity, but the sludge decreased the thermal conductivity values from $0.17 \pm 0.00$ to $0.16 \pm 0.00$ W m$^{-1}$ K$^{-1}$ under the air-dried condition (Table 4).

**Table 4.** (W m$^{-1}$ K$^{-1}$) of *Retisol* samples watered with the 0, 20, or 40% (0%, 20%, and 40% treatments, accordingly) of secondary pulp and paper sludge under the condition of saturated water content or field capacity and in air-dried soil.

| Sludge Dose, % | Saturated Water Content | Field Capacity | Air-Dried Soil |
|---|---|---|---|
| 0 | $0.92 \pm 0.01$ b | $0.83 \pm 0.04$ b | $0.17 \pm 0.00$ a |
| 20 | $0.92 \pm 0.05$ ab | $0.85 \pm 0.02$ b | $0.17 \pm 0.00$ a |
| 40 | $0.98 \pm 0.01$ a | $0.92 \pm 0.01$ a | $0.16 \pm 0.00$ b |

Different letters show significant differences between the means.

## 4. Discussion

In this study, we sought to test how the watering of plants with solutions of the secondary pulp and paper mill sludge affects the growth of *L. sativa* and the water-physical properties of *Retisol* on which these plants were grown. The effect of the sludge on the plant and soil was quite strong for most of the studied parameters. The sludge application enhanced the plant productivity by increasing the number of leaves and their LMA parameters (Table 1). This result is consistent with the early finding that secondary paper mill sludge can improve plant growth and productivity [19]. However, our study showed that sludge not only affects plant biomass accumulation but also biomass allocation. This is due to the fact that the sludge-mediated increase in biomass accumulation was more pronounced in the shoots than in the roots (Table 1). Similar to earlier findings showing that increased plant growth appears to be a result of improved soil properties [17,40], we attribute the increased growth in lettuce to the improved physical properties of *Retisol* on which the plants were grown. Increased pore space, saturated water content, and thermal conductivity of *Retisol* under the condition of secondary pulp and paper mill sludge application can be responsible for the improvement in the physiological state and productivity of *L. sativa* plants.

The results of this study indicate that no significant differences were found between the untreated and treated soil in the particle and water-stable microaggregate size distribution (Tables 2 and 3). This can be explained by the fact that the applied sludge solution contained many large particles that did not penetrate the pore space of the soil and formed a dark film on the soil surface. Although not significantly different, the mean content of the water-stable and non-water-stable aggregates changed when the secondary sludge was added to the soil. The sludge-mediated increase in the content of the silt-sized particles (0.002–0.05 mm) may be the result of the interactions of soil particles and sludge particles with the emergence of new stable organo–mineral complexes (Table 2). In addition, the higher content of the sand-sized water-stable aggregate fraction (>0.05 mm) in the sludge-treated soil, compared to both the secondary sludge used in this study and untreated soil, confirms the possibility of interaction between soil particles and sludge particles, which leads to the formation of new structures (Table 3). Thus, an increase in the content of the sand fraction reflects the formation of water-resistant particles of larger size under the condition of secondary sludge application.

The modification of soil structural properties can be responsible for changes in soil hydraulic conductivity and water retention, as has been found for soils with biochar [41–43]. For sludges, Chow et al. [22] showed significant increases in both soil porosity and moisture-holding properties. The study on water-resistant aggregates suggests that the binding of soil particles with the formation of secondary structures causes modifications of the pore space with a larger proportion of macropores compared to micropores [22]. The increase in the percentage of water-stable aggregates >0.05 mm found in our study should lead to a change in the structure of the pore space and the distribution of pores by size. No significant impact of sludge on available water capacity (pF > 2.5) was found (Figure 1).

Probably, the increase in the proportion of fine fractions did not lead to any significant change in the available water capacity due to their insignificant manifestation at the sludge dosages used in this study, the duration of the experiment, and the type of sludge used. However, the water retention curves reflect the increase in the saturated water content under the condition of secondary sludge application (Figure 1). Such an effect is likely to be related to the increased proportion of large drainage pores, which is confirmed by the shift in the air-entry values of the sludge-treated soil samples towards lower pF values. It cannot be ruled out that the increase in the proportion of large pores was facilitated by the increase in root growth.

The accumulation of sludge particles on the soil surface, as indicated by a dark surface color, can affect the saturated hydraulic conductivity of the soil. The sludge film on the surface can serve as a barrier to water penetration into the soil. This is probably the reason for the wide range of saturated hydraulic conductivity values in the samples treated with the 20% sludge solution (Figure 2). This treatment, on the one hand, led to an increase in the proportion of large pores, and on the other hand, it did not cause the formation of such a continuous and thick film on the soil surface as when treated with the 40% solution. The soil surface properties under the secondary sludge application with watering of plants should be studied in the future.

The sludge-mediated increase in the water content in the range from field capacity to saturated water capacity may be responsible for the increase in the effective thermal conductivity of the soil under study. Soil thermal properties play a very important role in land surface processes, but are strongly affected by a complicated complex of factors, including the geometry of the pore space and the solid matrix, composition, shape, and configuration of the various components in the medium [44]. Of particular importance is the influence of the water content in the medium on thermal conductivity [45]. The presence of water increases the thermal conductivity of the medium and, with an increase in water content, its effect can be significantly enhanced [43]. He et al. [46] analyzed the soil thermal conductivity in silt-organic soil mixtures and concluded that the thermal conductivity of the samples increases with the rise of soil water saturation. With an increase in the saturated water content, the contact area between soil and water particles increases, thereby enhancing heat transfer [46]. Moreover, the content of soil organic matter is considered one of the major factors affecting thermal conductivity in the soils. Previous studies have shown a negative correlation between thermal conductivity and soil organic matter content [46,47]. Pulp and paper mill sludges are rich in organic matter [15,28], although the increase in soil organic matter resulting from the addition of paper mill sludge depends on the doses and frequency of application, as well as sludge composition [12]. Typically, the pulp and paper mill sludge content ranges from 40% for primary sludge to 50% for secondary sludge [15]. Despite the entry of organic matter into the soil under the sludge application, it negatively affects the thermal conductivity of the studied soils only at low soil water values (Table 4). At high soil moisture, thermal conductivity depended mainly on the water content, which was higher in the sludge-treated soil. (Figure 1). Thus, given the positive relationship between the saturated water content and thermal conductivity, as well as the increase in thermal conductivity in sludge-treated soil, the effect of increasing the saturated water content seems to be more pronounced than the effect of increasing the content of organic matter.

## 5. Conclusions

This study showed a significant effect of secondary pulp and paper mill sludge, an organic-rich industrial material, on the growth parameters of *L. sativa*, as well as on most of the physical properties of *Retisol*. The sludge application caused an increase in plant growth with a change in plant biomass allocation due to the accumulation of more mass in the shoots than in the roots. The sludge-mediated increase in lettuce productivity was closely related to the soil's physical properties. Although the secondary sludge application when watering plants resulted in partial sludge accumulation on the soil surface, sludge

particles penetrated into the soil could interact with soil solids to form stable organo-mineral particles, which contributed to an increase in the proportion of large drainage pores and saturated water content. However, further study of the effect of secondary sludges on soil physical properties is necessary using alternative techniques for sludge application into soils. The secondary sludge decreased the effective thermal conductivity of *Retisol* in an air-dried state but increased the heat transfer in wet soil, which may be associated with the sludge-mediated increase in the soil water content under near-saturated conditions. Taken together, the results indicate a positive effect from the use of secondary pulp and paper sludge when watering plants; improving the water and thermal properties of the soil could indirectly affect the stimulation of plant growth.

**Author Contributions:** Conceptualization, M.B. and E.I.; methodology, M.B.; software, M.B. and E.I.; validation, M.B. and E.I.; formal analysis, M.B. and E.I.; investigation, M.B. and E.I.; resources, E.I.; data curation, M.B.; writing—M.B. and E.I.; writing—review and editing, E.I.; visualization, M.B. and E.I.; project administration, E.I.; funding acquisition, E.I. All authors have read and agreed to the published version of the manuscript.

**Funding:** This research was funded by RSF, grant number 22-16-00145. The study of plant growth parameters under secondary sludge application was supported by the Ministry of Science and Higher Education of the Russian Federation, grant number FMEN-2022-0004.

**Data Availability Statement:** Not applicable.

**Conflicts of Interest:** The authors declare no conflict of interest. The funders had no role in the design of the study; in the collection, analyses, or interpretation of data; in the writing of the manuscript; or in the decision to publish the results.

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
