# Peer review of "Physical Properties of Retisol under Secondary Pulp and Paper Sludge Application"

_land, doi:10.3390/land12112022_

Round 1

Reviewer 1 Report

Comments and Suggestions for Authors

The paper deals with an important issue related to sustainable soil management and global food security. It is generally well organized and written. Some other points to consider are:

1) Please add research meaning to this article.

2) The values of getted results are missing in the abstract. Please, add it.

3) In the literature section authors write about sludge effect on chemical soil properties, but they don't mention about the effect of this type sludge usage on physical properties, what is the main goal of this article.

4) It is unclear why authors selected to carry out the research even in Retisol? What are the main problems in this type of soil? In my opinion, this is needed to mention, because Retisol has a specific conditions comparing it with others types of soil.

5) instead section "2.1. Soil collection and plant growth condition" it is better to write "Pot experiment installation".

6) In my opinion, it would be better if authors will rite separate paragraph about the preparation of pot experiment mentioning how much soil was added to the plastic pots, how it was prepared. Now everything is mixed, it is a little bit difficult to understand. Also, it is needed to write the scheme of experiment.

7) in all Results section, it is necessary to write values of getted results, not just to write it is decreased or increased.

8) all abbreviations must be explained when they are mentioned for the first time in the text.

9) Figure 2 must be corrected, the writings in Russian should be deleted and the values of Y axis should be added.

10) the comparison with the results obtained by other authors is insufficient. the explanations and possible reasons related to the getted results are missing.

11) Please provide a more clear and concise problem statement in your discussion. A problem statement should be a concise and concrete summary of the research problem.

12) After discussion, limitations and future prospects need to be pointed out.

13) After the References, the template of MDPI articles should be deleted.

Author Response

First we would like thank the Reviewer for attentive revision of our manuscript and constructive comments and suggestions.

Let me indicate the modifications made in the manuscript in the light of Reviewer’s comments.

1) Please add research meaning to this article.

Thank You. Done.

2) The values of getted results are missing in the abstract. Please, add it.

Thank You, we added main results in the Abstract.

‘The sludge application enhanced plant growth with an increase in biomass accumulation of 21 and 53%, respectively, for 20 and 40% sludge treatment. When the sludge dose was increased from 0 to 40%, number of leaves increased by 25%, and values of leaf mass per area increased by 42%.’

And

‘Secondary sludge application led to an increase in the saturated water content from 0.50 to 0.56 cm3 cm-3. The 40% sludge solution increased soil thermal conductivity from 0.92 to 0.98 W m-1 â—¦C-1 under saturated water content and from 0.83 to 0.92 W m-1 â—¦C-1 under field capacity’.

3) In the literature section authors write about sludge effect on chemical soil properties, but they don't mention about the effect of this type sludge usage on physical properties, what is the main goal of this article.

In the section Introduction, we added sludge application effect on the soil physical properties:

‘Pulp and paper mill sludges application led, as a rule, to a decrease in soil bulk density [21] and to an increase in the ratio of macropores to micropores [22]. Sludge-mediated macroaggregate formation may be responsible for improving soil water infiltration and storage [22], thereby providing increased water availability to plants and consequently increased plant productivity and yield.’

4) It is unclear why authors selected to carry out the research even in Retisol? What are the main problems in this type of soil? In my opinion, this is needed to mention, because Retisol has a specific conditions comparing it with others types of soil.

Yes, You are right.

In the section Introduction, we added next:

‘This type of soil is often found in northern latitudes and is used for agricultural purposes. However, Retisol is characterized by poor fertility and unfavorable physical properties. Increasing the productivity of plants cultivated on this type of soil is possible only with the use of technologies that improve soil properties, including physical ones. Our hypothesis is that the secondary sludge application can improve soil properties by increasing its water holding capacity and thermal conductivity. Because sludge-mediated changes in soil properties are expected to increase plant productivity, in this study we assess whether secondary sludge application would improve growth parameters of lettuce (Lactuca sativa L.) plants cultivated on this type of soil.’

5) instead section "2.1. Soil collection and plant growth condition" it is better to write "Pot experiment installation".

Thank You, done.

6) In my opinion, it would be better if authors will rite separate paragraph about the preparation of pot experiment mentioning how much soil was added to the plastic pots, how it was prepared. Now everything is mixed, it is a little bit difficult to understand. Also, it is needed to write the scheme of experiment.

You are right. Thank You. We added in the section “M and M’ next:

‘After the fertilizing, the soil was well-watered and incubated under 21–23 â—¦C of air temperature and 70–80% of the maximum soil water holding capacity for 14 days, and then parked into plastic 0.80 L pots with a soil bulk density of approximately 1.4 g dry mass cm−3. Two days before sowing, the soil in each pot was moistened until saturated water content.’

7) in all Results section, it is necessary to write values of getted results, not just to write it is decreased or increased.

Thank You. Throughout the Results section, the results were added in the text.

8) all abbreviations must be explained when they are mentioned for the first time in the text.

You are right. An explanation of the abbreviation pF is added to the “Materials and Methods” section.

9) Figure 2 must be corrected, the writings in Russian should be deleted and the values of Y axis should be added.

Thank You. Done.

10) the comparison with the results obtained by other authors is insufficient. the explanations and possible reasons related to the getted results are missing.

Thank You. The results of the study are discussed in the section Discussion.

11) Please provide a more clear and concise problem statement in your discussion. A problem statement should be a concise and concrete summary of the research problem.

12) After discussion, limitations and future prospects need to be pointed out.

Thank You. According to your comments, we did change the text and added next:

‘Although the secondary sludge application when watering plants resulted in the sludge accumulation on the soil surface, the results showed that sludge particles interacted with soil solids to form stable organo-mineral particles with the increase in the proportion of large drainage pores and soil water-holding capacity. However, further study of the effect of secondary sludges on soil physical properties is necessary using alternative techniques for sludge application into soils.’

13) After the References, the template of MDPI articles should be deleted.

Done. Thank You.  

All changes in the text are highlighted in yellow.

Once again, let us thank You for your thorough review of our manuscript.

Reviewer 2 Report

Comments and Suggestions for Authors

I think the topic addressed in the paper was of great economic and environmental value, and will add knowledge to science. Great job. 

I think the paper was well written and the information was well organized. 1. What is the main question addressed by the research? The objectives of the study was to investigate the short-term effects of secondary sludge (pulp and paper mill) application on lettuce and the quality of retisol soil in the Northeast of Russia. Plant growth parameters and soil descriptive parameters were compared between several levels of application. Positive effects on desirable parameters were found, supporting the perceived short-term benefits that could be obtained from secondary sludge application on lettuce crops grown on retisol soil in the NE Russia. 2. Do you consider the topic original or relevant in the field? Does it address a specific gap in the field? As described and documented in the introduction, I believe this study to be original and relevant to the field of sustainability, circular economy (nutrient cycling), and soil health. 3. What does it add to the subject area compared with other published material? It augments the literature information about the use of secondary sludges on agricultural fields. 4. What specific improvements should the authors consider regarding the methodology? What further controls should be considered? Methodology wise, I think the study was well designed and conducted. However, I think looking at the effects of the sludges application on the same field / plot as the application rate is increased from 0% to higher rate could have added more information, instead of comparing the rate across different fields / plot. 5. Are the conclusions consistent with the evidence and arguments presented and do they address the main question posed? Yes 6. Are the references appropriate? Yes 7. Please include any additional comments on the tables and figures. No comments. I think the tables and figures were well designed, and all the information pieces are provided for them to be self-explanatory.

Author Response

Thank You!

All changes in the text are highlighted in yellow.

Best regards!

Reviewer 3 Report

Comments and Suggestions for Authors

Reviewer’s Comment

General Comments

It is good to know about your study on the “Physical properties of retisol under secondary pulp and paper sludge application”. Using a sustainable alternative to boost soil nutrients for crop production is a good idea but not a recent one. The flow and readability are not so good. I would suggest some major corrections to be made. Below are some specific comments.

Specific comments

Introduction: Pg 2; Lines 67-70:  Could you go further by giving examples of the benefits of the use of primary sludge and the potential examples of the benefits of secondary sludge?

In your introduction, you did not include any information about the previously studied chemical parameters. I suggest you include this information.

Aim: Pg 2; Lines 84-86:This study aimed to evaluate a short-term effect of secondary pulp and paper mill sludge on the characteristic of Retisol, such as soil structure and aggregation, hydraulic and thermal conductivity, soil water holding capacity, saturated water content, field capacity, available water capacity.” What do you mean by a “short-term effect”? Please explain what you mean.

Also, there are some terminologies you need to define in your introduction section. Terms such as “Retisol, hydraulic connectivity, and thermal properties”. Include a definition for a better understanding.

Subsection 3.4: Saturated hydraulic conductivity; needs more explanation. I absolutely do not understand what you mean. What are the treatments used? I would suggest you also add this information to your figure. Please add more details. Your Figure 2 has a language other than English. Could you correct that? Also, change the title of Figure 2 to a more understandable title.

Line 267: Remove one “that”

Lines 273-274: “Improved chemical parameters provide increased…” I thought you were supposed to report your findings on the physical properties of retisol… but your report now includes chemical properties. I suggest you explain what you mean.

Lines 277-279: “Results of this study indicate that no significant differences were found between the untreated and treated soil in the particle and water-stable microaggregate size distribution” Based on your results, your study lacks originality. If there is a novelty in your findings, I suggest you explicitly state it. If there is none, I suggest you state what your future study plans are, as regards achieving the aim of your study, or give recommendations for further studies.

Reference Section: Your reference section is divided, which should not be so. Please look at it and make the necessary corrections. Also, I would suggest that you remove the author's guide details from your manuscript. That should not be left in your manuscript.

Overall, I suggest you do a major revision to your manuscript.

Comments on the Quality of English Language

The quality of the English Language needs improvement.

Author Response

First we would like thank the Reviewer for attentive revision of our manuscript and constructive comments and suggestions.

Let me indicate the modifications made in the manuscript in the light of Reviewer’s comments.

Introduction: Pg 2; Lines 67-70:  Could you go further by giving examples of the benefits of the use of primary sludge and the potential examples of the benefits of secondary sludge?

Thank You. In the Introduction section we added next:

‘Pulp and paper mill sludges application led, as a rule, to a decrease in soil bulk density [21] and to an increase in the ratio of macropores to micropores [22]. Sludge-mediated macroaggregate formation may be responsible for improving soil water infiltration and storage [22], thereby providing increased water availability to plants and consequently increased plant productivity and yield.’

In your introduction, you did not include any information about the previously studied chemical parameters. I suggest you include this information.

The effect of secondary sludges on chemical parameters of soils and plants are shown now more clearly in the Introduction section. Thank You.

Aim: Pg 2; Lines 84-86: “This study aimed to evaluate a short-term effect of secondary pulp and paper mill sludge on the characteristic of Retisol, such as soil structure and aggregation, hydraulic and thermal conductivity, soil water holding capacity, saturated water content, field capacity, available water capacity.” What do you mean by a “short-term effect”? Please explain what you mean.

You are right, this term should be explained. We did that with adding in the text next:

‘ Despite the decomposition of organic matter and loss of soil carbon, the effects of the pulp and paper mill sludges on soil carbon stores were noted even several years after the sludges were added to soils [21, 31] . Moreover, Zibilske et al. [21] showed that the initial (short-term) effect during the first year may be lower than five years after sludge application, likely due to the time required for organic matter to decompose.’

Also, there are some terminologies you need to define in your introduction section. Terms such as “Retisol, hydraulic connectivity, and thermal properties”. Include a definition for a better understanding.

Thank You. According to your comment, we have made changes to the text. We added:

‘This type of soil is often found in northern latitudes and is used for agricultural purposes. However, Retisol is characterized by poor fertility and unfavorable physical properties. Increasing the productivity of plants cultivated on this type of soil is possible only with the use of technologies that improve soil properties, including physical ones. Our hypothesis is that the secondary sludge application can improve soil properties of Retisol by increasing its water holding capacity and thermal conductivity.’

‘The saturated hydraulic conductivity (Ks) reflects the ability of soils to pass water through pore spaces under saturated condition. The Ks values of undisturbed soil were measured in the laboratory using the constant head method [35]. Before starting the measurement, the soil cores were completely saturated. Throughout the experiment, a constant layer of water on the soil surface was maintained at a level of 1 cm. The time of sequential drainage of every 10 ml of water was measured and Darcy’s equation [36] was used to calculate the Ks values.’

Also we used ‘thermal conductivity’ instead of ‘thermal properties’.

Subsection 3.4: Saturated hydraulic conductivity; needs more explanation. I absolutely do not understand what you mean. What are the treatments used? I would suggest you also add this information to your figure. Please add more details. Your Figure 2 has a language other than English. Could you correct that? Also, change the title of Figure 2 to a more understandable title.

According to You comment the subsection ‘Saturated hydraulic conductivity’ was rewritten and Figure 2 was changed.

Line 267: Remove one “that”

Thank You! Done.

Lines 273-274: “Improved chemical parameters provide increased…” I thought you were supposed to report your findings on the physical properties of retisol… but your report now includes chemical properties. I suggest you explain what you mean.

Yes, this sentence was not needed in the text, so we deleted it.

Lines 277-279: “Results of this study indicate that no significant differences were found between the untreated and treated soil in the particle and water-stable microaggregate size distribution” Based on your results, your study lacks originality. If there is a novelty in your findings, I suggest you explicitly state it. If there is none, I suggest you state what your future study plans are, as regards achieving the aim of your study, or give recommendations for further studies.

Thank You. We have rewritten the Conclusion section according to your comment. We added:

‘Although the secondary sludge application when watering plants resulted in the sludge accumulation on the soil surface, the results showed that sludge particles interacted with soil solids to form stable organo-mineral particles with the increase in the proportion of large drainage pores and soil water-holding capacity. However, further study of the effect of secondary sludges on soil physical properties is necessary using alternative techniques for sludge application into soils.’

Reference Section: Your reference section is divided, which should not be so. Please look at it and make the necessary corrections. Also, I would suggest that you remove the author's guide details from your manuscript. That should not be left in your manuscript.

 Thank You. Done.

All changes in the text are highlighted in yellow.

Once again, let me thank You for your thorough review of our manuscript.

Round 2

Reviewer 1 Report

Comments and Suggestions for Authors

Authors considered many of the submitted comments, qualitatively revised the article. Some minor corrections before accepting the article for publication:

1) Please, check the units correctness (line 23).

2) Soil type should be written in Italic.

3) Some grammatical, word choice and sentence structure corrections are recommended

Author Response

Once again we thank the Reviewer.

Let us indicate the modifications made in the manuscript in the light of Reviewer’s comments.

  • Please, check the units correctness (line 23).

Thank You. We have checked the units. They are correct.

  • Soil type should be written in Italic.

Done. Thank You.

  • Some grammatical, word choice and sentence structure corrections are recommended

We checked the text and made corrections.

All changes to the text are highlighted in yellow.

Reviewer 3 Report

Comments and Suggestions for Authors

The authors have done a major revision to their manuscript.

Author Response

Once again we thank the Reviewer.